# Bidimensional and Contrast-Enhanced Ultrasonography of the Spleen in Dogs Affected by Leishmaniosis

**DOI:** 10.3390/ani11051437

**Published:** 2021-05-17

**Authors:** Massimo De Majo, Giulia Donato, Marisa Masucci, Cyndi Mangano, Maria Flaminia Persichetti, Luigi Liotta, Giuseppe Mazzullo, Rosanna Visalli, Marco Quartuccio, Nicola Maria Iannelli, Santo Cristarella, Maria Grazia Pennisi

**Affiliations:** 1Department of Veterinary Sciences, University of Messina, 98168 Messina, Italy; mdemajo@unime.it (M.D.M.); donatogiulia92@gmail.com (G.D.); cyndi_m@hotmail.it (C.M.); mfpersichetti@gmail.com (M.F.P.); luigi.liotta@unime.it (L.L.); giuseppe.mazzullo@unime.it (G.M.); mquartuccio@unime.it (M.Q.); nicola_iannelli@libero.it (N.M.I.); scristarella@unime.it (S.C.); mariagrazia.pennisi@unime.it (M.G.P.); 2Biogene, Veterinary Diagnostic Center, 95127 Catania, Italy; Biogene@biogene.it

**Keywords:** dog, contrast-enhanced ultrasonography, spleen, leishmaniosis

## Abstract

**Simple Summary:**

Canine leishmaniosis is responsible for pathological changes in the spleen. The main features detectable from ultrasound examination are splenomegaly and diffuse alterations of the echostructure. The study aimed to highlight whether these ultrasound changes are related to the severity of the disease or to a modification of splenic microvascularization that can be detected in vivo through contrast-enhanced ultrasonography. Bidimensional ultrasonography showed that splenomegaly and diffuse parenchymal changes were positively correlated with the severity of the disease, so they could be of prognostic value. Contrast-enhanced ultrasonography showed that a persistent heterogeneous distribution pattern appeared only in spleens with diffuse echostructure alterations, and quantitative parameters regarding volume and velocity of flow in three regions of interest did not show any differences between affected and control dogs. Diffuse splenic microvascular modifications evidenced by contrast-enhanced ultrasonography were reported for the first time in dogs with canine leishmaniosis.

**Abstract:**

Canine leishmaniosis (CanL) is responsible for splenic pathological changes. The main features detectable from ultrasound examination are splenomegaly and diffuse alterations of the echostructure. The study aimed to highlight whether these ultrasound changes are related to the severity of the disease or to a modification of splenic microvascularization that can be detected in vivo through contrast-enhanced ultrasonography (CEUS). Twenty-five adult dogs tested for CanL were enrolled in this prospective, controlled study and staged according to LeishVet guidelines. Bidimensional ultrasonography revealed that splenomegaly was seen in 50% of the affected dogs, and diffuse parenchymal changes were seen in more than 60% of dogs with splenomegaly, showing a positive correlation with severity of the disease; therefore, splenomegaly could be of prognostic significance. CEUS showed that a persistent heterogeneous distribution pattern appeared only in spleens with diffuse echostructure alterations. The evaluation of quantitative CEUS parameters regarding the volume and velocity of flow in three regions of interest did not show differences between affected and control dogs. Diffuse spleen microvascular modifications evidenced by CEUS were reported for the first time in dogs with CanL. In endemic areas, CanL could be included in the differential diagnoses list when detecting splenic alterations in dogs.

## 1. Introduction

Canine leishmaniosis (CanL) is a globally distributed zoonosis responsible for potentially serious fatal diseases in both dogs and humans [1,2,3,4]. *Leishmania infantum* is the most important *Leishmania* spp. worldwide and the main reported species in Europe [4]. Female sand flies from the genera *Phlebotomus* in the Old World are responsible for the vector-borne transmission of *Leishmania*, although other non-vectorial ways (transplacental, venereal, and by blood transfusion) can occur [2]. The interaction between the parasite and the host immune system determines the clinical manifestations and outcome of infection in CanL. The development of clinical signs in dogs is mainly related to the adaptive immune response with a predominant T helper 2 (Th2) and compromised T helper 1 (Th1) response that evolves into immune exhaustion [1,5]. The most commonly reported clinical signs in CanL are skin lesions, generalized lymphadenomegaly, weight loss, splenomegaly, exercise intolerance, muscular atrophy, decreased appetite, lethargy, polyuria and polydipsia, ocular lesions, epistaxis, onychogryphosis, lameness, vomiting, and diarrhea. Chronic renal failure is a severe result of disease progression and the main cause of mortality due to CanL [1,2,6,7]. 

The spleen has a role of protection against several blood-borne pathogens, and it is also one of the main organs involved in visceral leishmaniasis [8]. Histopathological splenic architecture disruption is characterized by the disorganization of white pulp [9,10], with reductions in size and the number of lymphoid follicles and of periarteriolar sheaths, furthermore a low number of CD4+ lymphocytes in the red pulp is reported [11].

In leishmaniotic dogs with splenic enlargement, abdominal ultrasonography has revealed that enlargement is mostly associated with a diffuse hypoechoic or coarse hypoechoic parenchymal pattern; however, a honey-comb aspect with a micronodular hypoechoic diffuse pattern in two dogs and a heterogeneous structure in another one were reported [12]. 

The scanning electron microscopy of the features of the spleens of dogs with *L. infantum* infection showed significant modifications of the vascular bed with a lack of sinusoids of the white pulp and a remarkable development of pulpar veins. These severe microvascular modifications were found in all spleens of all examined dogs and considered not related to an IFAT (immunofluorescence antibody test) titer [13]. Contrast-enhanced ultrasonography (CEUS) increases the scattered signals from vessels and, particularly, the microvasculature because of the use of a contrast agent. The CEUS analysis of the spleen has been proposed in some studies to establish normal perfusion patterns and to discriminate between malignant versus benign splenic lesions, although some findings about nodular lymphoid hyperplasia have made further investigation necessary [14,15,16,17,18]. 

The aims of the present study were (i) to investigate splenic abnormalities via the bidimensional ultrasonography in dogs affected by CanL, (ii) to perform CEUS exams to understand whether the reported anatomical vascular alterations were responsible for an in vivo modification of the diffusion of the contrast agent, and (iii) to evaluate the relationship between ultrasonographic splenic abnormalities, the clinical stage of CanL, and the spleen parasite load.

## 2. Materials and Methods

### 2.1. Study Design, Inclusion Criteria, and Exclusion Criteria

This prospective, analytical, observational, controlled study was performed from February 2018 to December 2019, and it included dogs visiting the Veterinary Teaching Hospital (University of Messina, Messina) (38°13′57″ N, 15°33′03″ E) and the Clinica Veterinaria Camagna of Reggio Calabria (38°06′54″ N, 15°39′25″ E). Written informed owner consent was obtained before dog enrollment. Inclusion criteria were adult age and antibody and/or PCR positivity to *L. infantum*. Dogs <1 year of age were enrolled only if they had been exposed to sand fly vectors for at least along one whole transmission season (April–October). Similarly, control dogs recruited for the study were adults, but they were antibody-negative and PCR-negative for *L. infantum*. Dogs with the ultrasonographic assessment of the focal lesions (nodule or mass) of the spleen or the cytological diagnosis of pyogranulomatous splenitis, lymphoma, or another malignant neoplasia were excluded.

### 2.2. Clinical Data Collection

For each dog, signalment, clinical history, and physical examination data were registered. Blood pressure (BP) was evaluated in a quiet area, before any other procedures, following ACVIM guidelines [19]. BP was measured with a high definition oscillometry device (Vet HDO) and classified according to IRIS guidelines [20]. Though all observed clinical signs were recorded, special attention was paid to identifying the most commonly reported ones in dogs with CanL [2].

### 2.3. Sampling Procedures

Five milliliters of blood were taken from each dog: one milliliter was placed into a K_2_EDTA (ethylenediaminetetraacetic acid) tube and used for complete blood count (CBC) (within 24 h), and the rest were stored until molecular investigations at −20 °C. Blood smears and serum were obtained by the remaining blood (about 4 mL). Blood serum was used for biochemical investigations and then stored until further use for serological tests at −20 °C. Urine samples (about 5 mL) were collected by cystocentesis or free catch. Urinalysis was performed within two hours after collection. The supernatant was used for the evaluation of urine protein to creatinine ratio (UPC) within 24 h after collection. In addition to swabs from any skin or mucosal lesions, oral, conjunctival, and auricular swabs were collected using individual sterile cotton swabs rolled, respectively, on the mucosal surface of fauces, lower conjunctival fornix, and the ear canal. FNA (fine needle aspirates) were taken from enlarged lymph nodes. The syringe and the needle used for lymph node aspiration and swabs were aseptically saved at −20 °C until analyzed for the detection of *L. infantum* DNA. After sedation with butorphanol (0.11 mg/kg IM), the percutaneous ultrasound-guided FNA of the splenic parenchyma was performed. Smears of the spleen aspirates were realized for cytological evaluation, and the syringe and the needle used were saved at −20 °C until further use for molecular investigations.

### 2.4. Staging of Leishmaniosis according to LeishVet Guidelines

Based on clinical and clinicopathological findings, enrolled dogs with CanL were clinically staged according to the classification system (I, IIa, IIb, III, and IV) established by LeishVet guidelines [1]. This staging system relates any of the four clinical stages with expected prognosis and therefore with the degree of disease severity. Antibody- and/or PCR-positive dogs to *L. infantum* with no clinical and clinico-pathological abnormalities were considered to be infected clinically healthy (ICH).

### 2.5. Clinicopathological Evaluation

A laser hematology analyzer (IDEXX ProCyte Dx^®^ Hematology Analyzer, Idexx Laboratories, Westbrook, Maine, ME, USA) was used to perform the CBC. Dogs were considered anemic if their hemoglobin concentration was below the reference interval (<13.1 g/dL). May–Grünwald–Giemsa (MGG) staining was used to stain blood smears. Blood smears were examined to estimate platelet number and evaluate differential leucocytic count [21]. A biochemical profile including blood urea nitrogen (BUN), serum creatinine, total proteins, albumin, alanine aminotransferase (ALT), and aspartate aminotransferase (AST) was performed at Biogene laboratory (Catania, Italy). Urinalysis was performed via dipstick analysis (Combur 9 Test strips, Roche Diagnostics, Indianapolis, Indiana, USA). A VET360 refractometer (Reichert, Seefeld, Germany) was used to measure USG, and Kova glasstic slides (Kova International, Garden Grove, CA, USA) were used to perform the microscopic evaluation of urine sediment. The UPC was evaluated at Biogene laboratory (Catania, Italy) using 0.5 as a cut-off value for proteinuria; UPC values between ≥0.2 and ≤0.5 were considered as borderline proteinuria [20].

### 2.6. Detection of Antibodies against Leishmania and Other Vector-Borne Pathogens (VBPs) 

Anti-*L. infantum* antibodies (IgG) were detected by IFAT using *L. infantum* (strain MHOM/IT/80/IPT1) antigen slides produced by Centro di Referenza Nazionale Leishmaniosi (C.Re.Na.L., Palermo, Italy) and rabbit anti-dog IgG (anti-dog IgG-FITC, Sigma Aldrich, St. Louis, MO, USA). The manufacturer’s protocol was followed, and the end-point titer of the positive samples was determined by preparing PBS (phosphate-buffered Saline) serial two-fold dilutions of serum starting from a dilution of 1:80. The cut off dilution value for positivity was set at 1:80 [1]. The presence of serum IgG against *Rickettsia conorii*, *Ehrlichia canis*, *Anaplasma phagocytophilum*, and *Coxiella burnetii* antigens was tested by IFAT using commercial kits (Fuller Laboratories Fullerton, California, USA). For all serological tests, the manufacturer’s protocol was followed. Cut-off dilutions of 1:80 for *A. phagocytophilum*, 1:64 for *R. conorii*, 1:50 for *E. canis*, and 1:16 for *C. burnetii* were used.

### 2.7. DNA Extraction and Leishmania Real-Time PCR

DNA was extracted from 200 µL of K_2_EDTA blood samples; oral, conjunctival, and auricular swabs; swabs from lesions; and lymph node and spleen FNA aspirates using the PureLink Genomic DNA kit (Invitrogen, Carlsbad, CA, USA) according to the manufacturer’s instructions. At the end of the extraction procedure, DNA was eluted in 100 µL of a PureLink genomic elution buffer and stored at −20 °C until used. As described elsewhere, quantitative real-time PCR, targeting the constant region in the minicircle kinetoplast DNA (NCBI accession number AF291093), was performed [22].

### 2.8. Molecular Investigations for Other VBPs 

The DNA extracted from EDTA-blood and spleen FNA samples was used with by PCR to detect *R. conorii*, *E. canis*, *A. phagocytophilum*, *A. platys*, *Babesia canis*, and *C. burnetii*, as described previously [23,24,25,26,27,28,29,30] using an AB GeneAmp PCR system 2700 Cycle Sequencing (Applied Biosystems, Monza, Italy). Briefly, PCR was carried out in a final volume of 25 μL using 1 × 1.5 mM of Green GoTaq Reaction Buffer (5 × and 7.5 mM MgCl_2_), 10 mM dNTP Mix, 1.25 U GoTaq G2 DNA Polymerase (5 U/μL), 1 mM each primer, and 3 μL of DNA. The target amplified for each pathogen, the used primers, and the fragment length are shown in Table 1. The molecular investigation of *E. canis* and Omp *B Rickettsia* spp. gene was done with nested PCR, while the molecular investigation of the Omp *A. Rickettsia* spp. gene was done with semi-nested PCR. The thermal cycling profile depended on the amplified pathogen.

### 2.9. Spleen Ultrasonography Procedures

The B-mode examination of the spleen was performed in all dogs by the same operator (M.D.M.). A Mindray M9 ultrasound machine (Mindray Medical Italy, Trezzano sul Naviglio, Italy) and a linear probe (6.6–13.5 MHz) were used. Hairs over the ventral portion of the abdomen were clipped. Alcohol and coupling gel were applied to the skin. Standardized settings were used as much as possible for depth (4–5 cm), and overall gain, dynamic range, focal zone, and time-gain compensation were optimized. Spleen size was subjectively evaluated: enlargement was reported if the margins appeared rounded and the organ appeared caudally and in the right side of the abdomen [31]. Spleen echo-structure was classified, according to the following criteria, as normal when the contours were regular and smooth; the parenchyma was finely textured, homogeneous, and more echogenic than liver and the cortex of left kidney; there was a “moth-eaten appearance” corresponding to the presence of multiple, small, individual hypoechoic areas throughout the splenic parenchyma that gave a mottled echotexture; there was a “marbled appearance” when parenchyma was characterized by the presence of multiple, overlapping, and ill-defined areas with heterogeneous echogenicity [32]. The B-mode images were reviewed by two authors (M.D.M. and C.M.), and a consensus was reached for the classification of splenic parenchyma of the examined dogs. Color Power Doppler (CPD) examination, using a low pulsed repetition frequency and a proper gain, allowed us to evaluate the splenic vascularization and to assess the path of blood vessels without artifacts. CEUS was performed with a linear transducer at a frequency of 5.7 MHz, a mechanical index (MI) from 0.06 to 0.08, persistency off, and a wide dynamic range; a single focal zone was set beneath the splenic parenchyma. A dual live function with B-mode and contrast simultaneous images was used. The used contrast agent was a sulphur hexafluoride signal enhancer (SonoVue^®^, Bracco International, Milano, Italy), and it was prepared according to the manufacturer’s recommendations. An aliquot (0.03 mL/kg of body weight) of the contrast medium was rapidly infused via a 3-way valve and a 18- or 20-gauge catheter inserted in a cephalic vein; the syringe was always maintained in a horizontal position for injection. The catheter was then immediately flushed with 5 mL of a saline solution. Each dog received two bolus injections of contrast agent; the second injection of contrast medium was at least 7–8 min later the first. The timer was simultaneously activated with the inoculation of the contrast agent dose. Between the two injections, the microbubbles were destroyed with a high MI flash on the abdominal large vessels. During the first bolus injection, if necessary, total gain and time-gain compensation were optimized and not changed anymore during the second injection. The transducer was manually positioned, and the same splenic image was maintained during all the CEUS examinations. Good-quality video clips for 180 s, obtained during the second bolus examination, were stored and subsequently analyzed by the two operators (M.D.M. and C.M.). The splenic enhancement was classified normal if, after a rapid enhancement of the small splenic arteries (10–13 s) and a heterogeneous phase of enhancement of splenic tissue, it became homogeneous at peak enhancement with a slow decrease (wash out). Enhancement was heterogeneous if it never became homogeneous. The post-processing quantitative analysis of video-clips was performed using the contrast imaging quantitative analysis (QA) function of the ultrasound machine M9 without movie clip exportation. The contrast imaging QA system adopted a time–intensity analysis to obtain perfusion quantitative information of the volume and velocity of flow. For each dog, three regions of interest (ROIs) were manually drawn in the splenic parenchyma: two adjacent round ROIs (0.4 cm × 0.4 cm) located in a row and a larger oval ROI (0.6 cm × 1.0 cm) that encompassed the two smaller ROIs (Figure 1). The ROIs were located at a depth of approximately 1.0–2.0 cm and, during ROI selection, vascular structures were excluded. Calculated parameters included the following: GOF (goodness of fit), meaning the fit degree of the curve with a range of 0–1, where 1 means the fit curve fits the raw curve perfectly; BI (base intensity), the basic intensity of no contrast agent perfusion status; AT (arrival time), the time point where contrast intensity appears; TTP (time to peak), the time when the contrast intensity reaches peak value; PI (peak intensity), the contrast peak intensity; AS (ascending slope), the ascending slope of contrast, i.e., the slope between the start point of tissue perfusion to the peak; DT/2, the time when the intensity is half the value of the peak intensity; DS (descending slope), the descending slope of the curve; and the AUC (area under curve), which is used to calculate the area under the time–intensity curves during contrast perfusion.

### 2.10. Cytological Procedures

Three or more smears were prepared from spleen FNA, and they were stained with MGG. The following criteria were taken into account for each examined case: hemodilution, spindle cells (fibrocytes and endothelial cells), small mature lymphocytes, medium lymphocytes, lymphoblasts, occasional plasma cells, macrophages, mast cells, and platelet aggregates [33]. Further criteria were cellularity, homogeneous or mixed lymphoid population, and a more represented cytotype.

### 2.11. Statistical Analysis 

Based on ultrasound evaluation, spleen size was scored 0 in case of normal dimension and 1 when it was enlarged. Similarly, a score of 1 was given to normal spleen echogenicity, and a score of −1 was given in case of spleen echo-structure diffuse alterations [12] (modified). The data of quantitative parameters (CEUS) were analyzed by ANOVA using the GLM procedure of the SAS/STAT^®^ software (SAS Inst. Inc., Version 9.3, 2017; Cary, NC, USA) while considering the clinical stage and ROIs as variables. Differences in ROIs between clinical stages were not significant (*p* > 0.05); therefore, this variable was removed from the dataset. The separation of means was assessed by Tukey’s test, and differences were considered significant if *p* < 0.05. Results were reported as least squares means ± standard error of the mean. Pearson’s correlation coefficients (r) were used to measure the relationships between splenic alterations and enlargements to leishmaniosis stage, *L. infantum* quantitative PCR, and IFAT anti-*L. infantum* antibody titers. The test was performed by the SAS/STAT^®^ software (2017).

## 3. Results

### 3.1. Clinical Evaluation of Dogs and Staging of Leishmaniosis

Twenty-five dogs were included in the study. One Leishmania-positive dog was ICH, and 21 dogs were classified according to LeishVet clinical staging system (Table 2) [1]. The numerosity of dogs in the four different clinical stages was inhomogeneous, with about half the dogs staged with a moderate disease. Therefore, we grouped the data of the dogs in stages with a low numerosity and similar prognosis together [1]: ICH dogs with dogs with mild disease (stage 1) and severe disease dogs (stage 3) with those in stage 4 (very severe disease).

Three control dogs were enrolled. Most of the CanL-enrolled dogs were cross-breed dogs (*n* = 10) and less frequently purebred dogs represented by Dachshund (*n* = 3), Italian Pointing (*n* = 3), American Staffordshire Terrier (*n* = 2), English Setter (*n* = 2), Pointer (*n* = 1), and Poodle (*n* = 1). Ten dogs were females, and 12 dogs were males. Age ranged from seven to 120 months (mean ± SD = 58.14 ± 29.39). Control dogs were represented by two 18-month-old females and one 120-month-old male cross-breed dogs. *L. infantum* IFAT titers of CanL dogs ranged from 80 to 81,920 (25–75 percentile = 140–6400; median = 640). The positivity and parasite loads of PCR examined samples are reported in Table 3. 

The frequency of clinical signs and hematologic/biochemical alterations in leishmaniotic and in control dogs are reported in Table 4 and Table 5, respectively. 

Anemia was the most common hematologic abnormality of CanL dogs. Most anemic dogs had mild anemia (77.8%), while moderate and severe anemia were observed in just two dogs. Anemia was only regenerative in one dog with moderate anemia. No positivity to the analyzed VBPs was observed with PCR in K_2_EDTA samples and spleen aspirates. Some dogs were antibody-positive to *E. canis, R. conorii, A. phagocytophilum,* and *C. burnetii*, and VBP overall antibody prevalence is represented in Table 6.

### 3.2. B-Mode Ultrasonography

Splenic enlargement was found in 11/22 (50%) leishmaniotic patients. Overall, 7/11 (63.6%) had diffuse parenchyma abnormalities, which were always accompanied by an increased size (Figure 1). Parenchyma had a moth-eaten appearance in 4/7 and a marbled appearance in 3/7. In the remaining four, spleen was subjectively reported to be of a larger size than normal but with a normal echotexture. A positive correlation with the clinical stage of disease was detected for splenic enlargement (r = 0.634; *p* = 0.036) and diffuse parenchymal alterations (r = 0.655; *p* = 0.047). The correlation between ultrasonographic alteration and splenic quantitative PCR showed a low positive effect (r = 0.06; *p* = 0.779). Conversely, there was no correlation between spleen enlargement and echotexture with lymph node parasite load (r = 0.26; *p* = 0.573, r = −0.329; *p* = 0.4705, respectively) and IFAT title (r = 0.33, *p* = 0.1413, r = −0.38; *p* = 0.09, respectively). In none of the three dogs negative for *L. infantum* were splenic enlargement or abnormalities of echotexture detected. Upon CPD examination, flow signals were detected in both normal appearance and in moth-eaten and marbled spleens; under careful observation, CPD frames in moth-eaten spleens showed the absence of flow in the more visible hypoechoic foci (Figure 2).

### 3.3. CEUS

In leishmaniotic dogs without splenic changes detectable from ultrasound examination and control dogs, CEUS exams showed the normal rapid enhancement of splenic arteries (10–13 s) and a heterogeneous phase of the enhancement of splenic tissue that became homogeneous at the end of wash-in with a slow decay (Figure 3). 

Enhancement in the moth-eaten parenchymas had a heterogeneous distribution pattern with hypoenhancement/no enhancement areas in both the wash-in and wash-out phases; in marbled spleens, enhancement intensity was weak and heterogeneous (Figure 4 and Figure 5). 

The statistical analysis of the quantitative CEUS parameters regarding the volume and velocity of flow in the three traced ROIs in leishmaniotic and in control dogs showed no significant differences (Table 7). 

### 3.4. Cytology

In 18/22 leishmaniotic dogs and control ones, the observed smears were characterized by mild or moderate hemodilution, and cells were basically represented by a mixed lymphoid population, some spindle cells of trabecular origin, few non-degenerated neutrophils, occasional plasma cells, and macrophages. This picture referred to a normal splenic cytology. In four cases, an increased occurrence of intact or hypersegmented neutrophils along with few eosinophils was suggestive of acute splenitis. In one out of these four cases, numerous small bodies compatible with *Leishmania* amastigotes were seen free in the background and within some activated macrophages. No other splenic cytological abnormalities were observed.

## 4. Discussion

Honeycomb, Swiss-cheese-like, and moth-eaten are the metaphorical names used to describe the ultrasonographic finding of small hypoechoic nodules throughout the spleen, causing a spotted echotexture that is commonly seen not only in lymphoma and mast cell neoplasia but also in benign conditions (extramedullary hematopoiesis, lymphoid hyperplasia, and pyogranulomatous splenitis) in dogs and cats [34,35,36]. These above-mentioned metaphorical definitions are, however, synonyms. In this study, the B-mode ultrasound examination of the spleen revealed an increase in the size of the organ in half of the subjects with CanL, with diffuse echotexture abnormalities (moth-eaten or marbled) in about 30% of all affected subjects. Interestingly, parenchymal abnormalities were only seen in cases of enlarged spleens, with 63.6% of dogs with splenomegaly presenting one of the two observed patterns. Moreover, these alterations were significantly more frequent in those subjects with a more severe disease, and echotexture abnormalities showed a positive but not significant correlation with splenic *L. infantum* quantitative PCR. In a previous report, ultrasonographic splenomegaly and honeycomb or inhomogeneous patterns were reported in not dissimilar percentages of dogs; particularly, the authors stressed that the honeycomb appearance had never been previously associated specifically with CanL [12]. In our opinion, the moth-eaten or marbled echostructure aspects of the spleen are compatible with moderate–severe leishmaniosis, and they could be associated with the list of clinical and clinicopathological abnormalities detected in LeishVet stages II–IV. A limitation of our study was the lack of the histological confirmation of the ultrasonography patterns; however, we performed a clinical field study, and the experimental design did not provide for the performance of a splenic biopsy or the sacrifice of the animal after the investigation procedures. This was due to ethical issues involved in taking splenic biopsies from normal dogs and the legal prohibition of sacrificing pets with leishmaniosis in Italy, apart from cases with a poor prognosis associated with a bad quality of life. Thus, only splenic fine needle aspirations were included in the diagnostic evaluation to exclude pathological conditions compatible with the observed diffuse ultrasound changes of the spleen. The absence of a statistically significant correlation between parasite loads and the ultrasound aspect could also have been a consequence of a lower sensitivity of the FNA sampling method due to the risk of getting inadequate (hemodiluted) or unrepresentative samples. In fact, a study performed on spleens obtained after dog culling showed different parasite loads in different areas of the parenchyma [10]. Therefore, we speculate that ultrasonographic findings could reflect the histopathological alterations that have been reported in the spleen of *Leishmania*-infected dogs and that the disorganization and destruction of the splenic lymphoid tissue are associated with a more severe disease [37,38]. However, we intend to carry out histological examination of the spleens of dogs with leishmaniosis in the case of culling after an ultrasound examination in order to correlate the ultrasound images to the histopathological pictures.

CPD examination and CEUS were used to study splenic microvascularization. After the inoculation of the contrast agent, normal splenic parenchyma showed the rapid enhancement of the small splenic arteries, a heterogeneous phase of enhancement that became homogeneous at the end of the wash-in phase, and a slow wash-out [39]. Though the contrast medium did not have a late tissue phase in the dog’s spleen, the slow decay of enhancement would have been related to the accumulation in the sinusoids network [40]. A CPD examination showed no flow signal from small hypoechoic areas in the moth-eaten pattern of the spleens. In moth-eaten spleens, CEUS was used for the first time and showed, through a persistence of diffuse inhomogeneity with areas of hypoenhancement or absence of enhancement, that the alterations found in the B-mode images also led to a modification of the vascular architecture throughout the organ. A similar heterogeneous appearance and a widespread hypoenhancement were also found in the marbled spleens. Spleen microvascular architecture changes emerged in *L. infantum*-positive dogs from an ultrastructural study independently of their antibody titers. A marked scarcity of the sinusoidal system sheet that surrounds the central artery/arteriole of the white pulp, a huge development of pulp venules and veins, and the presence of a development of reticular fibers were reported [13]. Interestingly, we evidenced that an abnormal pattern of enhancement was only in spleens with B-mode ultrasonography alterations, and we can only speculate about these results. This in vivo study of vascular architecture obtained with CEUS was, however, not able to highlight changes in the splenic vascular pattern in all infected dogs other than what was reported with ultrastructural evaluation [13]. The statistical analysis of the CEUS quantitative parameters, relating to the speed and volume of splenic blood flow, did not show statistically significant differences in the examined ROIs. This was probably due to the inhomogeneity of splenic enhancement in leishmaniotic subjects that was not quantitatively detectable through the pixel analysis of different areas, even when comparing relatively small ROIs, because of the widespread alteration of the splenic vascularity of parenchyma. Substantial evidence supports that the spleen plays a key role in the immunopathology of VBP infections [41,42]. Splenomegaly might be due to a multiplication of organisms within circulating mononuclear cells and mononuclear phagocytic tissues of spleen and lymph nodes. Splenomegaly is a result of reactive lymphoid hyperplasia and concurrent extramedullary hematopoiesis [43]. In canine babesiosis and ehrlichiosis, the most common splenic ultrasonographic findings have been a diffuse heterogeneous hypoechoic pattern and generalized splenomegaly [44,45]. CEUS exams were performed in the spleens of dogs with subclinical ehrlichiosis, and a higher velocity of blood flow (lower wash-in time, peak enhancement time, and wash-out time) compared to data from healthy dogs examined in previous studies was seen [45]. In our study, molecular investigations of blood samples and splenic aspirates showed them to be negative for *R. conorii*, *B. canis*, *E. canis*, *A. phagocytophilum*, *A. platys*, and *C. burnetii*, so we can consider it unlikely that the ultrasound changes of the spleen could be traced to other VBP coinfections with *L. infantum*.

## 5. Conclusions

CEUS is of increasing use in diagnostic imaging in clinical settings, with ever increasing fields of application in small animal pathologies. Among them, CanL is frequently responsible for splenomegaly and sometimes for echotexture diffuse alterations. In this study, the results of ultrasonographic exams showed a correlation with more severe clinical stages according to the classification system established by LeishVet guidelines. The prognostic significance of echostructure changes could be confirmed by following up dogs after therapy. Diffuse spleen microvascular modifications seen in CanL after performing CEUS exams were reported for the first time in the present report; the obtained data must be taken into account when evaluating splenic alterations in dogs living in endemic areas.

## Figures and Tables

**Figure 1 animals-11-01437-f001:**
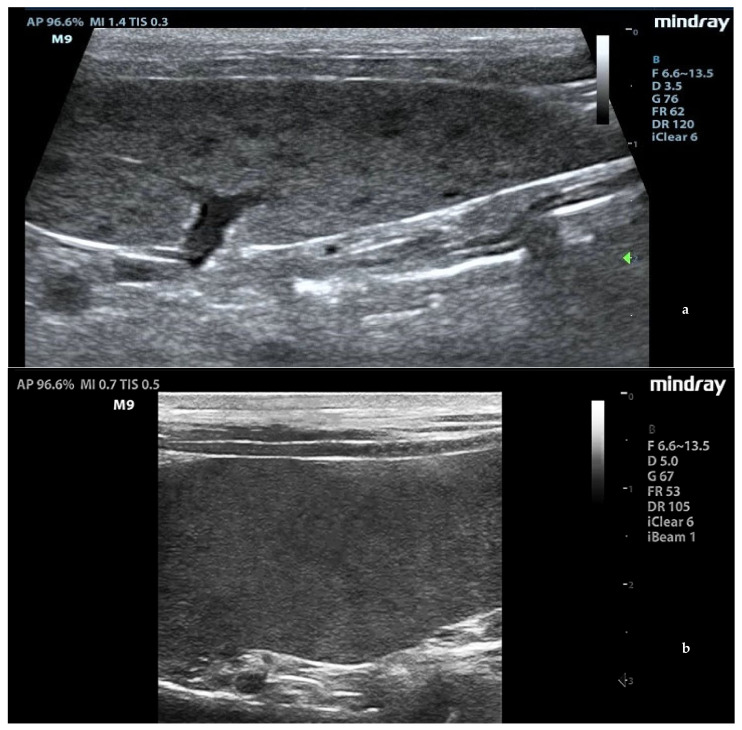
Ultrasound patterns of the spleen in *L. infantum*-infected dogs: (**a**) moth-eaten pattern and (**b**) marbled pattern. F: probe’s frequency; D: distance; G: gain; FR: frame rate; DR: dynamic range; AP: acoustic power; MI: mechanical index; TIS: tissue imaging specific; M9: ultrasound system. Green arrow: ultrasound focal point.

**Figure 2 animals-11-01437-f002:**
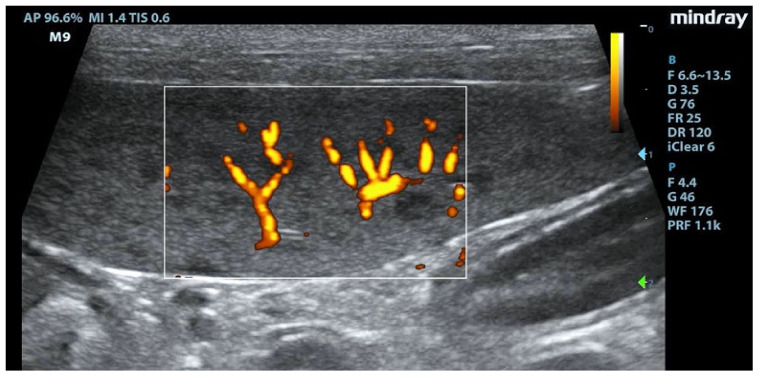
Absence of signal at Color Power Doppler (CPD) in hypoechoic areas of the moth-eaten pattern of the spleen. F: probe’s frequencies; D: distance; G: gain; FR: frame rate; DR: dynamic range; AP: acoustic power; MI: mechanichal index; TIS: tissue imaging specific; M9: ultrasound system. Blue and green arrows: focal points.

**Figure 3 animals-11-01437-f003:**
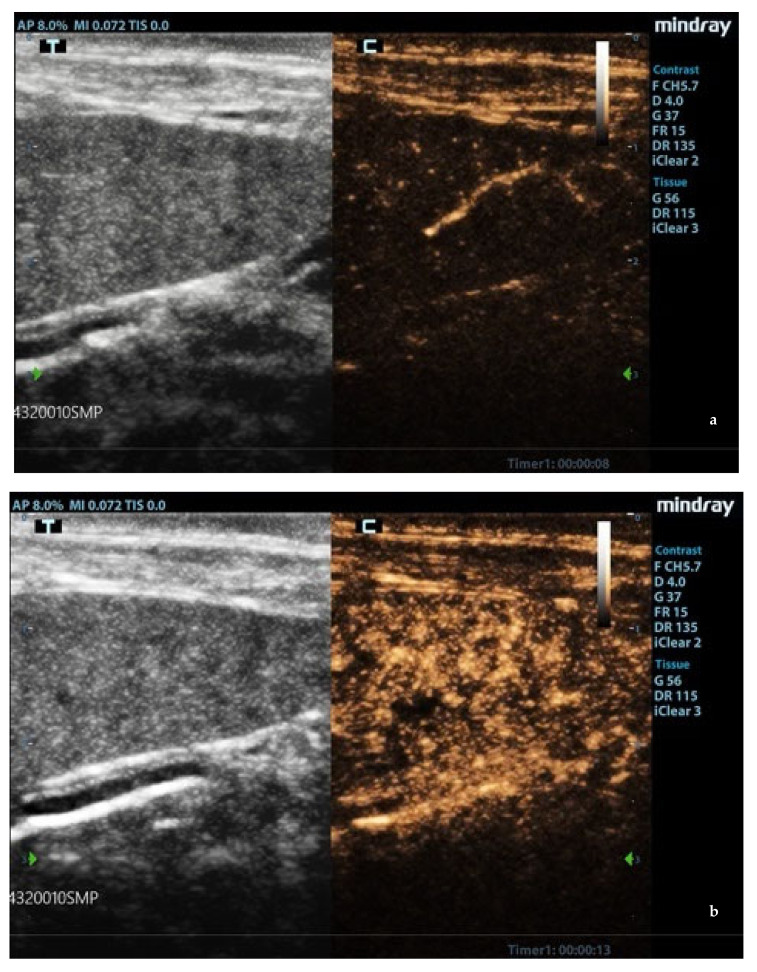
Normal spleen images acquired during CEUS (contrast-enhanced ultrasonography) exam. (**a**) Enhancement of the splenic arteries at 8 s after contrast injection; (**b**) beginning of heterogeneous phase of enhancement 10 s after contrast injection; (**c**) homogeneous enhancement at 60 s after contrast injection. F: probe’s frequencies; D: distance; G: gain; FR: frame rate; DR: dynamic range; AP: acoustic power; MI: mechanichal index; TIS: tissue imaging specific; M9: ultrasound system; T: tissue; C: contrast. Green arrows: focal points.

**Figure 4 animals-11-01437-f004:**
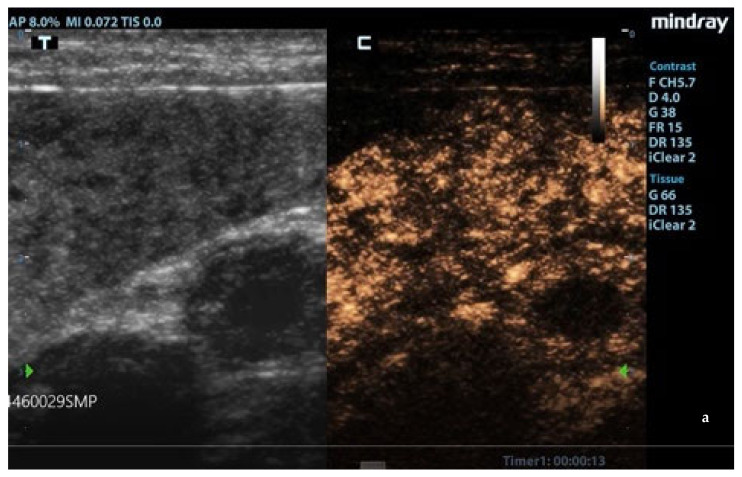
Moth-eaten spleen images acquired during CEUS exam. The heterogeneous enhancement of the parenchyma was persistently observed at: (**a**) 13 s, (**b**) 33 s, and (**c**) 60 s after contrast injection. F: probe’s frequencies; D: distance; G: gain; FR: frame rate; DR: dynamic range; AP: acoustic power; MI: mechanichal index; TIS: tissue imaging specific; M9: ultrasound system; T: tissue; C: contrast. Green arrows: focal points.

**Figure 5 animals-11-01437-f005:**
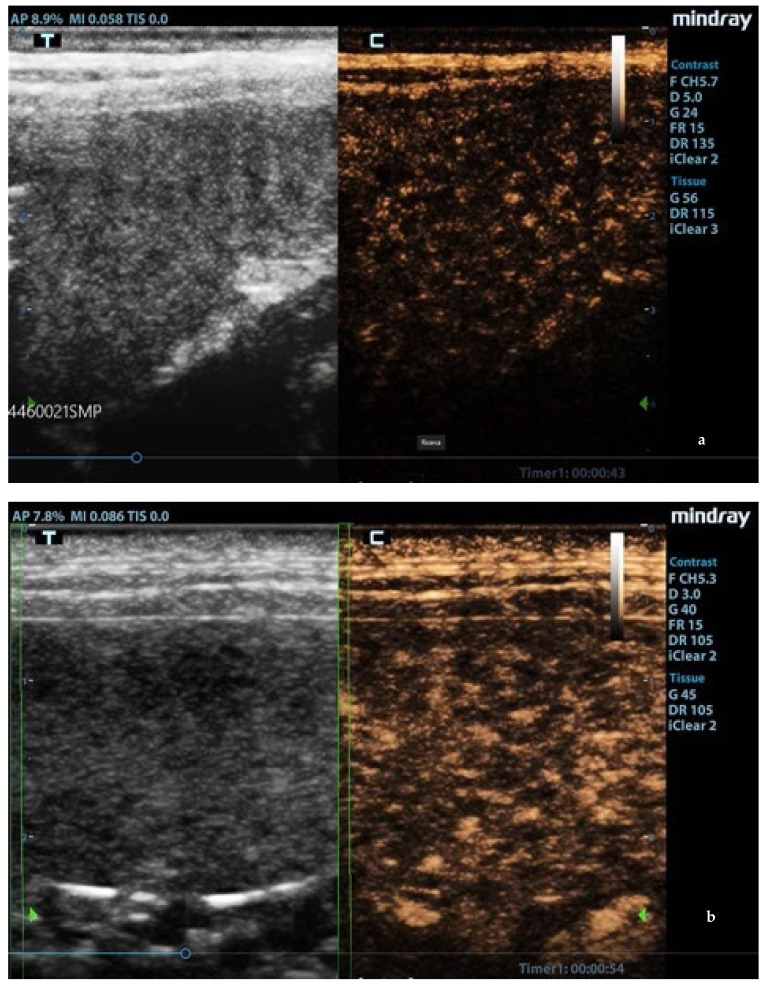
CEUS images at the end of the wash-in phase in (**a**) moth-eaten, (**b**) marbled (heterogeneous enhancement), and (**c**) normal (homogeneous enhancement) spleens. F: probe’s frequencies; D: distance; G: gain; FR: frame rate; DR: dynamic range; AP: acoustic power; MI: mechanichal index; TIS: tissue imaging specific; M9: ultrasound system; T: tissue; C: contrast. Green arrows: focal points.

**Table 1 animals-11-01437-t001:** Target amplified, primers used for pathogen detection, and fragment length.

Pathogen	Region Amplified	Primers (5′-3′)	Fragment Length (bp)	Reference
*Rickettsia* spp.	17KDa antigen	TZ15–19 5’-TTC TCA ATT CGG TAA GGG C-3’	246	[23]
		TZ16–20 5’-ATA TTG ACC AGT GCT ATT TC-3’		
*Rickettsia* spp.	Omp A	Rr190.70p ATGGCGAATATTTCTCCAAAA	532	[29]
		Rr190.701n GTTCCGTTAATGGCAGCATCT		
		Rr190.602n AGTGCAGCATTCGCTCCCCCT		
*Rickettsia* spp.	Omp B	rompB OF GTAACCGGAAGTAATCGTTTCGTAA	511/425	[27]
		rompB OR GCTTTATAACCAGCTAAACCACC		
		rompB SFG IF GTTTAATACGTGCTGCTAACCAA		
		rompB SFG/TG IR GGTTTGGCCCATATACCATAAG		
*E. canis*	16SrRNA	ECC AGAACGAACGCTGGCGGCAAGCC	480/390	[30]
		ECB CGTATTACCGCGGCTGCTGGCA		
		‘‘canis’’ CAATTATTTATAGCCTCTGGCTATAGGA		
		HE3 TATAGGTACCGTCATTATCTTCCCTAT		
*Anaplasma phagocytophilum*	msp4	MSP4AP5 5’-ATGAATTACAGAGAATTGCTTGTAGG-3’	849	[28]
		MSP4AP3 5’-TTAATTGAAAGCAAATCTTGCTCCTATG-3’		
*Anaplasma platys*	16SrRNA	PLATY-F 5’-AAG TCG AAC GGA TTT TTG TC-3′	~500	[26]
		PLATYS-R 5′-CTT TAA CTT ACC GAA CC-3′		
*Coxiella burnetiid*	htpB	Q5 (5′-GCG GGT GAT GGT ACC ACA ACA-3′)	501	[25]
		Q3 (5′-GGC AAT CAC CAA TAA GGG CCG-3′)		
		Q6 (5′-TT GCT GGA ATG AAC CCC A-3′)	325	
		Q4 (5′-TC AAG CTC CGC ACT CAT G-3′)		
*Babesia canis*	ssu-rDNA	PIRO-A 5′AATACCCAATCCTGACACAGGG 3’	~400	[24]
		PIRO-B 5’TTAAATACGAATGCCCCCAAC 3’		

**Table 2 animals-11-01437-t002:** Clinical classification of dogs into groups according to LeishVet guidelines [1]. ICH (infected clinically health).

Clinical Classification	Number of Dogs (%)
ICH	1 (4.5)
Stage I (mild disease)	5 (22.8)
Stage II (moderate disease)	
IIa	9 (40.9)
IIb	1 (4.5)
Stage III (severe disease)	2 (9.1)
Stage IV (very severe disease)	4 (18.2)

**Table 3 animals-11-01437-t003:** *L. infantum* molecular positivity and parasite loads in the examined samples.

Samples	Tested Samples	Positive Samples (%)	Parasite Load
Conjunctival swabs	52	1 (1.9)	85
Oral swabs	26	1 (3.8)	115
Auricular swabs	25	2 (8)	15–430
K_2_EDTA samples	26	4 (15.4)	30–70
Lymph node aspirates	15	9 (60)	110–6400
Spleen aspirates	26	8 (30.8)	10–18,000
Nodule aspirates	2	1 (50)	130

**Table 4 animals-11-01437-t004:** Frequency of clinical signs of dogs with canine leishmaniosis (CanL) and control dogs.

Clinical Signs	Number of Dogs (%)	
	Dogs with CanL	Control Dogs
Low Body Condition Score (BCS/9)	8 (36.4)	0
Low Muscle Condition Score (MCS/4)	5 (22.7)	0
Decreased appetite	1 (4.5)	0
Lethargy	0	0
Fever	0	0
Lymphadenomegaly	17 (77.3)	1 (33.3)
Local	15 (68.2)	0
Generalized	2 (9.1)	1 (33.3)
Skin lesionsNodular dermatitis	9 (40.9)3 (13.6)	0-
Ulcerative dermatitis	1 (4.5)	-
Squamous dermatitis	4 (18.2)	-
Alopecia	5 (22.7)	-
Splenomegaly	8 (36.4)	0
Epistaxis	0	0
Ocular lesions	6 (27.3)	0
Blepharoconjunctivitis	4 (18.2)	-
Conjunctival granulomas	0	-
Keratouveitis	2 (9.1)	-

**Table 5 animals-11-01437-t005:** Frequency of hematologic and biochemical alterations of dogs with canine leishmaniosis (CanL) and control dogs. * Thrombocytopenia was not confirmed on blood smears in one CanL dog and in one control dog. MCV (mean red blood cell volume), MCHC (mean corpuscular hemoglobin concentration), RDW (red blood cell distribution width), BUN (blood urea nitrogen), TP (total proteins), ALT (alanine aminotransferase), AST (aspartate aminotransferase), UPC (urine-specific gravity), WRI (within the reference interval).

Parameter (units)	High (%)		Low (%)		WRI (%)		Reference Interval
	Dogs with CanL	Control Dogs	Dogs with CanL	Control Dogs	Dogs with CanL	Control Dogs	
*Hematology*							
Red blood cells (M/µL)	2 (9.1)	1 (33.3)	6 (27.3)	0	14 (63.6)	2 (66.7)	5.65–8.87
Hematocrit (%)	1 (4.6)	0	7 (31.8)	0	14 (63.6)	3 (100)	37.3–61.7
Hemoglobin (g/dL)	1 (4.5)	0	9 (40.9)	1 (33.3)	12 (54.6)	2 (66.7)	13.1–20.5
MCV (fL)	0	0	4 (18.2)	0	18 (81.8)	3 (100)	61.6–73.5
MCHC (g/dL)	0	0	3 (13.6)	0	19 (86.4)	3 (100)	32.0–37.9
RDW (%)	0	0	0	0	22	3 (100)	13.6–21.7
Reticulocytes (K/µL)	1 (4.5)	1 (33.3)	0	0	21 (95.5)	2 (66.7)	10.0–110.0
White blood cells (K/µL)	1 (4.5)	0	0	0	21 (95.5)	3 (100)	5.05–16.76
Neutrophils (K/µL)	1 (4.5)	0	0	1 (33.3)	21 (95.5)	2 (66.7)	2.95–11.64
Lymphocytes (K/µL)	0	0	3 (13.6)	0	19 (86.4)	3 (100)	1.05–5.10
Monocytes (K/µL)	3 (13.6)	0	0	0	19 (86.4)	3 (100)	0.16–1.12
Eosinophils (K/µL)	7 (31.8)	1 (33.3)	2 (9.1)	0	13 (59.1)	2 (66.7)	0.06–1.23
Basophils (K/µL)	3 (13.6)	0	0	0	19 (86.4)	3 (100)	0.00–0.10
Platelets (K/µL)	0	0	5 * (22.7)	1 *(33.3)	17 (77.3)	2 (66.7)	148–484
*Biochemistry*							
BUN (mg/dL)	3 (13.6)	0	0	0	19 (86.4)	3 (100)	10–25
Creatinine (mg/dL)	1 (4.5)	0	0	0	21 (95.5)	3 (100)	<2
TP (g/dL)	2 (9.1)	1 (33.3)	1 (4.5)	0	19 (86.4)	2 (66.7)	5.5–7.8
Albumin (g/dL)	0	0	6 (27.3)	1 (33.3)	16 (72.7)	2 (66.7)	2.5–3.5
ALT (UI/L)	2 (9.1)	0	0	0	20 (90.9)	3 (100)	<100
AST (UI/L)	0	0	0	0	22	3 (100)	<90
UPC (no units)	7 (31.8)	0	0	0	15 (68.2)	3 (100)	<0.5

**Table 6 animals-11-01437-t006:** Vector-borne pathogen (VBP) overall antibody prevalence of dogs with canine leishmaniosis (CanL) and control dogs.

Pathogens	Seroreactive Dogs (%)	
	CanL	Control Dogs
*Ehrlichia canis*	3/22 (13.6)	1/3 (33.3)
*Rickettsia conorii*	13/22 (59.1)	3/3 (100)
*Anaplasma phagocytophilum*	5/21 (23.8)	2/3 (66.7)
*Coxiella burnetii*	2/22 (9.1)	1/3 (33.3)

**Table 7 animals-11-01437-t007:** Results from ANOVA for quantitative CEUS parameters in relations to ROI (region of interest) areas. GOF: goodness of fit; BI: base intensity; AT: arrival time; TTP: time to peak; PI: peak intensity; AS: ascending slope; DT/2: time when the intensity is half the value of the peak intensity; DS: descending slope; AUC: area under curve; and SEM: standard error of the mean.

	GOF	BI	AT	TTP	PI	AS	DT/2	DS	AUC
ROI1	0.83	15.62	2.27	31.02	21.33	0.18	109.72	−0.03	3303.61
ROI2	0.82	15.90	2.18	28.53	21.47	0.15	109.08	−0.03	3283.48
ROI3	0.90	15.87	1.30	30.28	21.44	0.14	116.12	−0.03	3335.92
SEM	0.05	0.21	0.08	1.71	2.65	0.15	2.55	0.09	18.38
*p*-value	0.06	0.48	0.08	0.24	0.49	0.30	0.16	0.12	0.38

## Data Availability

The datasets used and/or analyzed during the current study are available from the corresponding author on reasonable request.

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
