# Peer review of "Bidimensional and Contrast-Enhanced Ultrasonography of the Spleen in Dogs Affected by Leishmaniosis"

_animals, 2021, doi:10.3390/ani11051437_

Round 1

Reviewer 1 Report

In the manuscript entitled “Bidimensional and contrast-enhanced ultrasonography of the spleen in dogs affected by leishmaniosis” the authors discuss the characterization of bidimensional and contrast-enhanced ultrasonography of the spleen of leishmaniosis affected dogs. Below are the suggestions to improve the manuscript.

  1. How are spleen ultrasonography findings corroborated by other leishmania specific immune parameters or histopathology changes? The authors should explain in detail.
  2. The number of animals included is on the lower side especially in Stage IIb and Stage III categories. How can you determine statistically significant findings?
  3. Lines 48 & 49: The interaction between the parasite and the host immune system determines the clinical manifestations and outcome of infection in CanL.
  4. Lines 52 & 53: The most commonly reported clinical signs in CanL are skin lesions, generalized lymphadenomegaly, weight loss, muscular atrophy

Reviewer 2 Report

Reviewer comments for manuscript ID animals -1182600 entitled ‘Bidimensional and contrast-enhanced ultrasonography of the spleen in dogs affected by leishmaniosis’

General comments

Leishmaniasis is an important disease of public health significance worldwide. This is a valuable study in the ultrasonology of spleen of affected dogs along with its correlation with cytological changes. It is a novel study and diagnostic imaging of such an important disease will further increase the research interests to study this disease from an emphasis on diagnostic approaches. I congratulate the authors for this exhaustive work of high significance. Ultrasonographic procedures are very nicely described.

 I am impressed with the ultrasonology and cytology and its presentation. There are few corrections / suggestions pointed out. Conclusions section should include what further work the authors envisage in future studies.

Specific comments

Line 13: Please reword ‘splenic pathological changes’ as ‘pathological changes in the spleen’

Line 18: Please replace ‘a more severe disease’ with ‘severity of the disease’

Line 19: Please reword ‘it can take on a prognostic significance’ as ‘it can be of prognostic value’

Line 22: Please replace ‘spleen’ with ‘splenic’

Line 33: Please replace ‘a more severe disease’ with ‘severity of the disease’

Line 33: Please reframe ‘it can take on a’ as ‘it can be of prognostic significance’

Line 38: Please replace ‘should be’ with ‘could be’

Line 91: Please delete ‘preliminary’

Line 109: Please reframe ‘and stored at’ as ‘and the rest stored at ‘

Line 115: I am sorry I could not understand this word ‘auricolar’. Is it auricular?

Line 136: Please reword ‘and leukocyte differential count’ as ‘and to evaluate differential leucocytic count’

Line 150: Please delete ‘were used’

Line 264: The control group was restricted to three dogs only. Is it statistically viable for comparison with disease group having 25 dogs? Please clarify.

Lines 372-75: Please delete these lines and replace with ‘This was due to ethical issues involved in taking biopsies from normal dogs and legal prohibition on sacrificing pets for experimental research studies in Italy’

Reviewer 3 Report

Dear authors,

I appreciate you work, I think it is an interesting paper about a condition that is becoming increasingly prevalent in a variety of areas. This work describes an innovative aspect and presents important, even if preliminary, results. I know that it is not the aim of the present study, but do you also evaluate dogs after treatment? I have some clarifications to ask you.

Line 195 and 217: Why don’t you evaluate the intra- and the inter-observer agreement?

Line 295: What do you mean with “severity of disease2? Do yoo consider the classification based on LeishVet guidelines or based only on clinical signs?

Line 351: You evaluate the cytology of the spleen but you don’t discuss the results of FNA of lymph nodes. I can read the result only on table 3. Why?

Line 363: There is a non-significative correlation between quantitative PCR of the spleen an echotexture of the spleen. Do you consider a possible correlation between quantitative PCR of lymph nodes or of other samples different from spleen and US findings?

Line 377: An histological examination was not possible and I agree with your motivations. You consider only parasite loads of the spleen samples. Is it possible that results could be different if considering parasite loads of lymp node or congiuntival, nasal, skin swabs?

Line 427: I will add a statement about moth-eaten and marbled pattern to highlight the importance of this finding.
